# Clinical and Serological Follow-Up of 216 Patients with Hematological Malignancies after Vaccination with Pfizer-BioNT162b2 mRNA COVID-19 in a Real-World Study

**DOI:** 10.3390/vaccines11030493

**Published:** 2023-02-21

**Authors:** Jean-François Rossi, Emmanuel Bonnet, Christel Castelli, Marion Velensek, Emma Wisniewski, Sophie Heraud, Rania Boustany, Céleste David, Jérôme Dinet, Roland Sicard, Jean-Pierre Daures, Marion Bonifacy, Lysiane Mousset, Emmanuel Goffart

**Affiliations:** 1Institut du Cancer Avignon-Provence, Sainte Catherine, 84000 Avignon, France; 2Faculté de Médecine Montpellier, Université de Montpellier, 34094 Montpellier, France; 3Recherche Clinique Clinique Beau Soleil—Nouvelles Technologies, 34070 Montpellier, France; 4Thess Corporate Inc., 34070 Montpellier, France; 5Bioaxiome-Inovie, 84000 Avignon, France

**Keywords:** COVID-19, serological follow-up, hematological malignancies

## Abstract

Hematological malignancies (HMs) have heterogeneous serological responses after vaccination due to disease or treatment. The aim of this real-world study was to analyze it after Pfizer-BioNT162b2 mRNA vaccination in 216 patients followed up for 1 year. The first 43 patients had an initial follow-up by a telemedicine (TM) system with no major events reported. The anti-spike IgG antibodies were checked 3–4 weeks post-first vaccination and every 3–4 months, by two standard bioassays and a rapid serological test (RST). Vaccine boosts were given when the level was <7 BAU/mL. Patients who did not seroconvert after 3–4 doses received tixagevimab/cilgavimab (TC). Fifteen results were discordant between two standard bioassays. Good agreement was observed between the standard and RST in 97 samples. After two doses, 68% were seroconverted (median = 59 BAU/mL) with a median of 162 BAU/mL and 9 BAU/mL, respectively, in untreated and treated patients (*p* < 0.001), particularly for patients receiving rituximab. Patients with gammaglobulin levels < 5 g/L had reduced seroconversion compared to higher levels (*p* = 0.019). The median levels were 228 BAU/mL post-second dose if seroconverted post-first and second, or if seroconverted only post-second dose. A total of 68% of post-second dose negative patients were post-third dose positive. A total of 16% received TC, six with non-severe symptomatic COVID-19 within 15–40 days. Personalized serological follow-up should apply particularly to patients with HMs.

## 1. Introduction

Vaccination represents the best validated strategy against the SARS-CoV-2 pandemic, especially for high-risk patients, such as patients with hematological malignancies [1,2]. These patients have been described to develop a delayed or weak specific immune response after standard vaccination due to disease or treatment [3]. The specific immune response to SARS-CoV-2 has been described as heterogenous and generally reduced in patients with cancer and, in particular, hematological malignancies [4,5,6]. Recently, a prospective study was conducted in immunocompromised patients, including patients with hematological malignancies, to analyze the decrease in anti-spike (S) and anti-N antibodies after vaccination [6]. The decrease in the specific antibody response was observed for all patients and controls at 6 months; however, it was greater in patients with hematological malignancies than in the other groups of patients and controls [6]. Additionally, different parameters have been shown to reduce seroconversion levels after general vaccination, such as immune senescence, inflammaging and prior antigen exposure, especially to latent viruses such as cytomegalovirus (CMV) [7,8]. It was observed that the response to vaccination against COVID-19 in patients with hematological malignancies was also influenced by the type of therapy as shown by different cohort analyses [4,5,6,9,10].

One of the fundamental questions that arises when faced with these patients is to decide which patients should be revaccinated, when, how many boosts shoots, and which strategy to adopt in the absence of seroconversion to protect these patients from a severe infection which can also delay their specific treatment with its prognostic consequence. The recommendations for the systematic use of vaccine boosts in COVID-19 for the entire population or subgroups considered to be at risk is a non-personalized attitude and may increase toxicity or be useless for the objective set by achieving immune overstimulation which must be evaluated. A more reasonable medical policy could be based on patient follow-up both with a telemedicine application for tolerance during the vaccination period and longitudinal follow-up of seroconversion with rapid serological tests (RST) to control a more personalized vaccination follow-up. The aim of this study was to report a real-world medical follow-up and decision in a series of 216 patients with HMs vaccinated with Pfizer-BioNT162b2 mRNA COVID-19 at a single institution.

## 2. Material and Methods

### 2.1. Patient Population

Patients with hematological malignancies, and non-malignant hemopathy, were followed by the same clinician (JFR) at the Avignon-Provence Cancer Institute, with standard bio-clinical follow-up from 2021 January to June 2022. If necessary, patients received standard treatment for their disease as defined in agreement with French regulations including multidisciplinary consultation meetings. Due to the COVID-19 context, patients received at least two doses of the Pfizer-BNT162b1 vaccine intramuscularly as recommended. The question was to adapt and adopt a logical medical decision after these two first vaccinations in a real-world context. The organization and collection of data in this real-world study was approved by the Internal Ethics Committee on 21 December 2020. Patients gave their informed consent after having received from the clinician (JFR) a document containing information allowing the use of anonymous data collected in accordance with relevant guidelines and regulations, in particular, EU RGPD n°2016/679 relating to the protection of natural persons regarding the processing of personal data.

### 2.2. Monitoring of Post-Vaccination Tolerance by Telemedicine

According to the French authorities, and due to initial follow-up after new mRNA vaccines, early tolerance after the first two vaccinations was assessed in the 43 first patients who were vaccinated at the institute. This initial monitoring was carried out using Thess^®^ monitoring, a telemedicine system linking the patient to the healthcare of the institute for 24/24 h, 7 days, and developed by Thess Corporate, Montpellier France (www.thess-corp.fr, accessed on 17 February 2023).

### 2.3. Serological Follow-Up

In addition to the standard clinical and biological monitoring, the serological immune response (anti-spike, IgG and IgG + M antibodies, Ab) was analyzed in the serum, 3 to 4 weeks after the first dose and 4 to 8 weeks after the second dose and every 2–3 months, by SARS-CoV-2 IgG II Quant^®^ Assay with a positivity threshold of 50 U/mL corresponding to 7 BAU/mL (Abbott, Rungis, France). In the absence of a biological standard at the start of serological analyses, the central laboratory used a second bioassay carried out in another laboratory of the same group (LABOSUD/INOVIE, Montpellier, France). Thus, 182 samples were analyzed using Elecsys^®^ Anti-SARS-CoV-2 S (Roche Diagnostics, Meylan, France) with a positivity threshold of 0.8 BAU/mL, to be compared to the results of the Quant^®^ Assay. In a subgroup of 97 samples, semi-quantitative analysis was carried out using a rapid serological test (RST), from BIOSIS Healing (Beijing, China) and compared to the Quant^®^ Assay and the Elecsys^®^ Assay. For the entire study, the Quant^®^ Assay was used as a reference for patient follow-up. Depending on the results of the serological monitoring, a third dose was administered if the patient was not seroconverted or if the level of antibody was below 10 BAU/mL (Quant^®^ Assay, Abbott), in particular for those who had a reduced or very short serological response. For a fourth dose, if the patient had never been seroconverted, we favored the therapeutic use of anti-COVID-19 monoclonal antibodies.

### 2.4. Bio-Medical Follow-Up and Effectiveness

Among the patients who received three doses, those who remained negative or with a level lower than 10 BAU/mL without recent COVID-19, received administration of tixagevimab/cilgavimab (TC, Evusheld^®^, Astra Zeneca France, Courbevoie, France). Patients infected by SARS-CoV-2 received casirivimab-imdevimab (CI, Ronapreve^®^, Roche France, Nanterre, France).

### 2.5. Statistical Analysis

This is an observational study. The usual descriptive statistics were used. Categorical variables were expressed as numbers and percentages and continuous variables as medians and interquartiles. Comparisons of continuous variables between the groups were performed using the Student’s *t*-test. Categorical variables were compared between groups by χ^2^ or Fisher’s exact test. All statistical tests were performed as 0.05 two-sided. Statistical analysis was performed using R version 4.0.3 (Microsoft, Redmond, WA, USA).

## 3. Results

### 3.1. Patient Population

Among a population of 238 patients, 10 were excluded, 5 for an unknown date of COVID-19 and 5 having COVID-19 before the second dose of the vaccine. As shown in Table 1, 216 patients with hematological malignancies and 12 patients with non-malignant hemopathy were followed in this “real-world” analysis. All had negative PCR and no antibodies against SARS-CoV-2 when the first dose of the vaccine was administered. All patients received 2 doses, 146 of them received 3 doses and 14 patients received 4 doses. Among the patients with hematological malignancies, there were 125 patients with lymphoproliferative disorders, including 96 patients with lymphoma (89 with non-Hodgkin’s lymphoma, NHL, and 7 with Hodgkin’s disease, HD). There were 124 males and 104 females, and the median age was 73 (range 58–82). A total of 105 patients (46%) received concomitant therapy for their disease, including rituximab-based therapy (30 patients), daratumumab based-therapy (13 patients) or other therapy. A total of 123 patients (54%) had no therapy during this follow-up period.

### 3.2. Early Assessment of Tolerance Post-Vaccination Follow-Up by Telemedicine

Local pain at the injection site, generally transient, was mainly reported after the second dose. Only 4 out of 43 patients reported significant adverse events via telemedicine, followed by a medical call. The main adverse events included severe asthenia for 2 or more days, fever (>38 °C) for at least 2 days, headache or generalized pain. These systemic symptoms were more frequently observed after the second dose. No other major adverse effects were reported. The satisfaction index of the monitoring system was good with the impact of reassuring both patients and health staff reached.

### 3.3. Comparison between the Tests for Measuring SARS-CoV-2 Spike Antibodies

As shown in Table 2, the comparison of results between the Elecsys Quant^®^ Assay and the Quant^®^ Assay was significantly concordant, on 182 samples, with only 15 discordant interpretations of the results, including 12 patients negative with Elecsys Quant^®^ Assay and positive with Quant^®^ Assay, and 3 discordant in the opposite way (*p <* 0.001). As shown in Table 3, the comparison of the results obtained with the RST showed good agreement with the standard tests overall. Thus, 11 and 8 of the 52 RST negative patients were positive with Quant^®^ Assay and Elecsys Quant^®^ Assay, respectively. Moreover, there was a correct gradation of the results according to the different categories of neutralizing titers, making it possible to consider this RST as a good screening test.

### 3.4. Seroconversion Levels

As shown in Table 1, after the second dose of the vaccine, 64% of the 228 patients were seroconverted with a median level of antibody at 59 BAU/mL (range 0–1889), including 62% of the 216 patients with hematological malignancies (median 55 BAU/mL, range 0–896). A total of 79 patients had serological analysis both after the first and second dose of the vaccine. As seen in Figure 1, the median levels of anti-spike Ab were amplified after the second dose, with better amplification for lymphoma patients. Of these 79 patients, 58 were not seroconverted after the first dose, with only 24/58 seroconverted only after the second dose with a median anti-S antibody at 16 BAU/ML (range 1–270) compared to a median antibody at 1679 BAU/mL (range 549–5516) after the second dose for the 23 patients who seroconverted after the first dose. 

### 3.5. Factors Influencing the Seroconversion

As shown in Table 1, the median age was similar across all disease types, and seroconversion after two doses was not disease dependent. Different factors influenced the occurrence of seroconversion and its level. The median rates of anti-spike Ab were at 162 BAU/mL in untreated patients and 9 BAU/mL in treated patients (*p <* 0.001). This difference was particularly observed in the 30 patients receiving rituximab. After 2 doses, only 11 of them (38%) were seroconverted with a median level of the anti-spike Ab at 0.4 BAU/mL (range 0.1–20), significantly lower than observed in patients receiving daratumumab (n = 13) with a median level at 8 BAU/mL (range 1–41). The lymphocyte count at the time of vaccination also impacted seroconversion after the second dose, with a median of 113 BAU/mL for the group of patients with 1.5 to 4.0 Giga/L, 69 BAU/mL for those with 0.8 to 1.5 Giga/L and 2.8 BAU/mL for patients having less than 0.8 Giga/L. This effect was, however, related to treatment, with 63% of patients of the patients treated in the groups of patients having a lymphocyte count lower than 1.5 Giga/L vs. 38% in the group of patients with a normal lymphocyte count.

The CMV status based on the presence of anti-CMV IgG antibodies was analyzed at the same time as the dosage of anti-spike Ab after the second dose in 85 patients, including 61 with a positive CMV status. The median level of anti-spike Ab was not different between the negative and positive patients, respectively, 4.6 BAU/mL (range 0–39,634) and 5.7 BAU/mL (range 0–1066). Gammaglobulin levels (<5 g/L) were also associated with a significantly lower antibody anti-spike level, respectively, 85 BAU/mL in lower gammaglobulin levels (<5 g/L) vs. 1801 BAU/mL in patients having higher levels (>5 g/L) (*p =* 0.019) after two doses. A total of 118 patients had immunoglobulins (Ig) levels assayed, showing a good correlation between IgG and gammaglobulin levels.

All the patients who had no seroconversion after the second dose had a third dose. A total of 42% of them were not seroconverted after a third dose. For all patients having a third dose, the median level of the anti-spike Ab was 49 BAU/mL (range 1–544).

### 3.6. Bio-Medical Follow-Up and Efficacy

As shown in Figure 2, 14 patients who had not seroconverted after at least 3 doses received a fourth dose and a fifth dose for 3 of them. These patients were still not seroconverted and we offered them TC, with two additional patients who had very low anti-spike Ab titers (less than 10 BAU/mL). However, six of them experienced COVID-19 within 15–30 days of injection with mild symptoms. A total of 39 (17%) patients became infected with SARS-CoV-2 after the second dose of the vaccination within a median time of 322 days (range 9–517), of which 29 patients became infected after the third dose within a median time of 207 days (range 14–404). Only one of them had severe COVID-19 requiring hospitalization for 30 days, including 10 days in an intensive care unit. The remaining patients were symptomatic for a median of 4 days without hospitalization. A total of 7 of them had an anti-spike antibody dosage with a mean level of 207 BAU/mL (range 0–654) before infection and 297 BAU/mL (range 0.1–5714) at a median of 38 days after the infection. No patient in this series died of COVID-19.

## 4. Discussion

Following the 21st Century Cures Act passed in 2016, “real-world data and real-world evidence are playing an increasing role in health care decisions” [11,12]. Vaccination against SARS-CoV-2 is one of the best examples for generating real-world data due to the accelerated medical context that requires adaptation to more personalized medicine. Thus, one of the first questions asked was the tolerance to vaccination, which has been widely assessed for the normal population in prospective registration studies as well as by self-reporting of vaccination in a dedicated centralized registry and different government reports [13,14]. In patients with cancer, tolerance has only been reported in association with immunogenicity studies also based on self-reported analyses [15]. In a recent study of 93 cancer patients, local adverse drug reactions were recently reported more frequently after the first and second doses than after the third, while systemic symptoms were observed to be more frequent after the third dose [16]. However, patients with hematological malignancies generally had higher levels of immunosuppression, which may increase toxicity in the context of novel vaccines. This observation was also made in our survey of the first 43 patients vaccinated at the institute. To our knowledge, no analysis has been made using a simple self-questionnaire included in a telemedicine application. At the beginning of the vaccination campaign, when new side effects appeared especially for a new type of vaccine, a misunderstood message appeared in this context leading to a more complex acceptance of COVID-19 vaccines. We, therefore, used such a tool to promote better patient adherence to the new vaccine by using this monitoring tool to reassure patients and medical staff about the novelty of the vaccine. Indeed, vaccine hesitancy against COVID-19 is relatively evenly distributed across the population, closely related to identifying collective importance, but this was likely less observed in a population with cancer [17]. In our survey including 43 patients followed with the telemedicine application, there was no patient refusing vaccination and only one among the following patients. This process is planned to be developed in other vaccination campaigns, such as pneumococci or influenza, and in various other situations.

Protecting cancer patients has been a priority in vaccine policies including COVID-19. Among the first published studies mentioning the post-vaccination COVID-19 serological response, few patients with hematological malignancies were included in the different series [4,5,6,9,10]. The role of treatments and, in particular, rituximab has been confirmed in the same way as for other vaccines. Seroconversion has also been described as impaired in patients with auto-immune diseases treated with rituximab [18]. Detectable B cells (at least 40/mm^3^) and a long time since the last exposure to rituximab have been associated with the development of anti-SARS-CoV-2 spike protein antibodies after the booster vaccine [19,20]. Additional questions relate to the policy of vaccination against COVID-19, especially for patients with hematological malignancies who have shown a heterogeneous response to vaccination as mentioned in a recent meta-analysis and subgroup analyses [21]. According to the OnCovid registry, vaccination against SARS-CoV-2 was associated with a decrease in morbidity and mortality related to COVID-19 in cancer patients including 346 patients with hematological malignancies [22]. There was a difference in mortality rates at 14 and 28 days between fully vaccinated and unvaccinated patients, 5.5% vs. 20.7% (*p =* 0.0004) and 13.2% vs. 27.4%, respectively (*p =* 0.0028). Fully vaccinated patients had fewer sequelae than unvaccinated patients (6.7% vs. 17.2%, *p =* 0.0320) [22]. This suggests that seroconversion and presumably serological level must influence COVID-19 morbidity and mortality in cancer and reinforces the need to verify the specific immune response by monitoring this response by adapting the strategy on a personalized basis. In addition to the threat posed by acute morbidity and mortality due to COVID-19 in cancer patients, recent evidence highlights that the continuity of oncology care may be further disrupted by the long-term consequences of COVID-19, which affect approximately 15% of cancer patients recovering from the acute phase [23]. Vaccination recommendations have been made for the elderly with routine vaccinations every 6 months without serological monitoring [24]. For immunocompromised people, the recommendations are unclear. In particular the number of booster shots, and the timing of using an antibody treatment such as casirivimab-imdevimab (Ronapreve^®^) or tixagevimab/cilgavimab (Evusheld^®^) [25,26,27,28]. In the randomized TC PROVENT study, only 7% of the participants had a history of cancer [29]. A comprehensive review of the clinical experience of TC therapy was recently published, including 4 clinical trials and 38 real-world data, with a demonstration of the benefits of TC on pre-exposure prophylaxis of immunocompromised populations, including patients with B-cell depleting therapy [30]. The experience of TC administration has been particularly reported in two series of patients with hematological malignancies with less than 4% of patients infected with COVID-19, without severe forms of diseases [30,31]. Thus, our real-life study integrates a global medical strategy adapted to high-risk patients with heterogeneous seroconversion, including the use of TC with an infection rate of 17% but no severe form. Immunodeficiency due to disease or treatment is well known to negatively impact seroconversion after vaccination, as observed after influenza vaccine, including H1N1, and pneumococcal, particularly after allograft or chimeric antigen receptor (CAR) T-cell therapy [32,33]. The heterogenous humoral response in hematological malignancies was due to the alteration of the B-cell compartment that is usually observed after the therapies used in hematological malignancies [34,35]. The specific cellular response against SARS-CoV-2 has also been described as heterogeneous in such diseases and not always correlated with the humoral response [36]. This observation may complicate the interpretation of the specific immune protection of these patients. Thus, there are several clinical options, including routine vaccination every 6 months, monitoring of cellular and serological immune response for all patients or monitoring of serological response only. The first option is an epidemiological response based on the apparently reduced severity of COVID-19 in patients who have full vaccinations for cancer patients and no personalized medical follow-up. Monitoring humoral and cellular immune response is not possible due to cost, although there are new bioassays and biological tests to monitor both, including commercially available assays for T-cell response, but they are of little practical use [36,37]. Only RST monitoring of the humoral response is simple, giving a semiquantitative response sufficient for tracking as we observed.

While third-dose boosters are effective for most people with cancer, increasing protection against the coronavirus, their effectiveness is heterogenous and lower than in the general population [38]. Many cancer patients will remain at increased risk of coronavirus infections even after three doses. This is probably due to a lack of immune efficacy in some patients that must be identified to protect them with neutralizing antibodies. Our study suggests that if patients had no serological response after two vaccines, it is probably best to use such therapies. The use of such neutralizing antibodies does not fully protect against COVID-19 but may likely reduce viral load to limit disease severity as we have observed.

In conclusion, due to the heterogeneity of seroprotection against COVID-19 in hematological malignancies, there is a need for personalized dynamic monitoring in the context of immune protection after vaccination, especially against COVID-19 and probably in different clinical situations such as standard vaccination for high-risk people. The ability to use RST makes this possibility more available.

## Figures and Tables

**Figure 1 vaccines-11-00493-f001:**
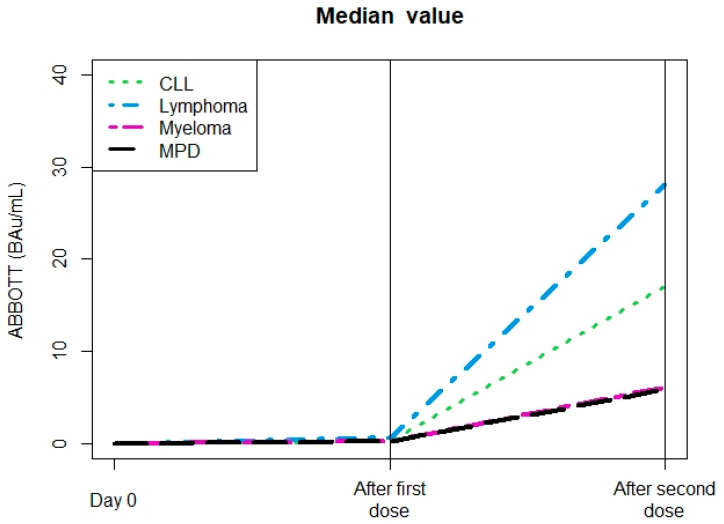
Median levels of anti-spike antibodies: comparison between the first and second doses. Abbreviations: CLL: Chronic lymphocytic leukemia; MPD: myeloproliferative disorders.

**Figure 2 vaccines-11-00493-f002:**
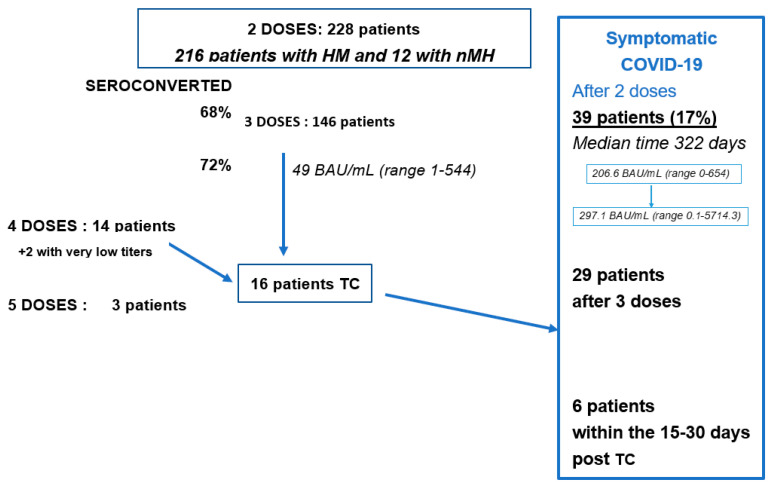
Summary of follow-up including patients who had symptomatic COVID-19. Abbreviations: HM: hematological malignancies; nMH: non-malignant hemopathies; TC: tixagevimab/cilgavimab (EVUSHELD^®^).

**Table 1 vaccines-11-00493-t001:** Patients’ characteristics for the different groups of diseases, number of patients seroconverted and median (range) levels of the anti-spike anti-antibodies, significantly different between patients with hematological malignancies (HMs) and patients with non-malignant hemopathy. Abbreviations: MGUS: monoclonal gammopathy of undetermined significance; CLL: chronic lymphocytic leukemia; MPD: myeloproliferative disorders.

	Lymphoma n = 96	Myeloma n = 33	MGUS n = 27	CLL n = 27	MPD n = 33	Others n = 12	Total n = 228
**Age**	71	75	73	72	77	71	**73**
Median (range) at the second dose	63–77	72–80	66–77	66–76	66–82	58–95	**58–82**
**Sex** M	54 (56%)	13 (39%)	15 (56%)	18 (67%)	19 (58%)	5 (42%)	**124 (54%)**
**Treatment** Yes	43 (55%)	26 (79%)	4 (15%) *	6 (22%)	21 (64%)	5 (41%)	**105 (46%)**
Daratumumab		7	4 *				
Rituximab	30						
**Nb of pts/dose**							
Only 2	26 (27%)	9 (27%)	4 (15%)	8 (30%)	17 (52%)	6 (50%)	**70 (30%)**
Only 3	67 (70%)	20 (61%)	23 (85%)	15 (56%)	13 (39%)	6 (50%)	**144 (63%)**
4 doses	3	4	0	4	3	0	**14 (6%)**
**Seroconverted** after the second dose	57 (66%)	19 (61%)	20 (91%)	13 (62%)	24 (80%)	14 (100%)	**14 (63%)**
Missing data	10	2	5	6		3	
**Levels of Anti-S Ab Median (IQR)** after the second dose (BAU/mL)	39 0–274	15 1–68	204 70–896	121 1–266	50 12–425	217 16–1889	**59** **3–319**

** p* < 0.001.

**Table 2 vaccines-11-00493-t002:** Concordance of the seroconversion after two doses between Elecsys Quant^®^ Assay (Abbott) and Quant^®^ Assay (Roche) on 182 samples, showing a significant correlation.

Seroconverted After Two Doses		Quant^®^ Assay (Roche)			
		No	Yes	Total	
**Elecsys Quant^®^ Assay (Abbott)**	**No**	**42 (23%)**	**12 (6.6%)**	54 (30%)	*p* < 0.001
	**Yes**	**3 (1.6%)**	**125 (69%)**	128 (70%)	
**Total**		**45 (25%)**	**137 (75%)**	182 (100%)	

**Table 3 vaccines-11-00493-t003:** Concordance of the seroconversion after two doses between a rapid serological test (RST), Elecsys Quant^®^ Assay (Abbott) and the Quant^®^ Assay (Roche), with a gradation of the reference titers (neutralization test 50%, 60, 120 and 300 U).

97 Samples RST.		Quant^®^ Assay (Mean, Range BAU/mL)	Elecsys^®^ Assay (Mean, Range BAU/mL)
**Negative 52/97 (54%)**		11+ (8.9, 5.8–15.3)	8+ (8.1, 1.4–24.3)
Positive 45/97 (46%)		160.7 (0.34–2291.0)	232.5 (0–>2432)
**Reference TITER (Neutralization test 50%)**			
**60**	17/45 (38%)	23.5 (0.34–66.7)	46.1 (2.3–220.9)
**120**	7/45 (16%)	54.1 (19.5–142.3)	84.9 (8.6–271.4)
**300**	21/45 (46%)	407.5 (44.5–2291.0)	465.8 (0–>2432)

## Data Availability

All the data generated and analyzed during the current study is hosted on the secure server of the Institute of Cancer Avignon-Provence belonging to the UNICANCER (French Cancer Centers) network in a nominative manner according to the European and the CNIL (Commission Nationale de l’Informatique et des Libertés) rules, and on the server of the Department de Recherche Clinique Beau Soleil—Nouvelles Technologies, Montpellier, France, directed by Professor Jean-Pierre Daures. All data are available from the corresponding author upon reasonable request. The questionnaire for telemedicine monitoring is available if requested.

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
