# Peer review of "Clinical and Serological Follow-Up of 216 Patients with Hematological Malignancies after Vaccination with Pfizer-BioNT162b2 mRNA COVID-19 in a Real-World Study"

_vaccines, 2023, doi:10.3390/vaccines11030493_

Round 1
Reviewer 1 Report (Previous Reviewer 1)
None.
Author Response
/
Reviewer 2 Report (Previous Reviewer 2)
.-Persist Mismatches between text and table 1
-In the text, it is said that the median age for the 228 patients was 73 years (range 62-82), but in table 1 appears as 73 (63-82). Please, correct and explain.
-Persist error in Table 1
-there is a “*” for the number 4. There is no legend to explain what means this “*”. It appears the same as the previous version.
-Table 2 is partially updated
The authors reply: “Table 2 represents the concordance of the positivity and negativity between the 2 tests. We modified the Table and the title for better understanding”
Which data correspond to what test (Elecsys Quant® Assay (Abbott) 134 and Quant® Assay (Roche))? This information should appear in the table. In the reply, the authors add the information, but not in the manuscript.
-Moreover, the numbers are wrong in the title of table 2: “11/52 patients and 8/11 were positive, respectively with Quant 138 Assay and Elecsys Assay,…”
It is not 8/11, it should be 8/52. Again a mistake in the numbers
-Lines 145-148: “79 patients had serological analysis both after the first and second dose of vaccine. As seen in Fig.2, the median levels of anti-spike Ab were amplified after the second dose, with better amplification for CLL patients”
Better compared with who? In figure 2, the best response seems to be for lymphoma patients, better than CLL.
-In figure 2, the scale goes only up to 40 BAU/ml, but in the text it is said: “Of these 79 patients, 58 were not seroconverted after the first dose, with only 24/58 seroconverted after the second dose with a median anti-S antibody at 16 BAU/ML (range 1-270) compared to a median anti-body at 1679 BAU/mL (range 549-5516) after the 2nd dose for the 23 patients who seroconverted after the first dose.” Figure 2 doesn't fit with the response seen in the 23 patients with a median of 1679 BAU/ml (the figure goes until 40 BAU/ml). Please, again, explain and correct.
-Lines 235-237: “To our knowledge, only 52 patients with hematological malignancies received TC, with 3.8% of patients with COVID-236 19 with a follow-up of 79 days (25).”
This statement is far away from reality. Many more oncohematologic patients have received tixagevimab/cilgavimab. Read, for example, a recent review of the topic (Akinosoglou K. Tixagevimab/Cilgavimab in SARS-CoV-2 Prophylaxis and Therapy: A Comprehensive Review of Clinical Experience. Viruses. 2023;15(1):118).
Author Response
Responses to comments from reviewer 2:
In the text, it is said that the median age for the 228 patients was 73 years (range 62-82), but in table 1 appears as 73 (63-82). Please, correct and explain.
response: Median age and range was re-calculated 73 (range 58-82) for the 228 patients and harmonized with Table 1.
there is a “*” for the number 4. There is no legend to explain what means this “*”. It appears the same as the previous version.
response: the word doses was added.
The authors reply: “Table 2 represents the concordance of the positivity and negativity between the 2 tests.
response : We modified the Table and the title for better understanding
we add in the text the sentence, "including 12 patients negative with Elecsys Quant® Assay and positive with Quant® Assay, and 3 discordant in the opposite way "
Moreover, the numbers are wrong in the title of table 2: “11/52 patients and 8/11 were positive, respectively with Quant 138 Assay and Elecsys Assay,…”
It is not 8/11, it should be 8/52. Again a mistake in the numbers
response: We agree in Table 3 the presentation of the data was not correct and we change it and added the following sentence in the text:
"Thus, 11 and 8 of the 52 RST negative patients were positive with Quant® Assay and Elecsys Quant® Assay respectively. Moreover, there was a correct gradation of the results according to the different categories of neutralizing titers, making it possible to consider this RST as a good screening test. "
Lines 145-148: “79 patients had serological analysis both after the first and second dose of vaccine. As seen in Fig.2, the median levels of anti-spike Ab were amplified after the second dose, with better amplification for CLL patients” Better compared with who? In figure 2, the best response seems to be for lymphoma patients, better than CLL.
response: we agree this figure 2 showed a comparison of the medians between different diseases between response after the first and the second dose, with the best response obtained in lymphoma patients
We changed the paragraph entitled "seronconversion levels"
-In figure 2, the scale goes only up to 40 BAU/ml, but in the text it is said: “Of these 79 patients, 58 were not seroconverted after the first dose, with only 24/58 seroconverted after the second dose with a median anti-S antibody at 16 BAU/ML (range 1-270) compared to a median anti-body at 1679 BAU/mL (range 549-5516) after the 2nd dose for the 23 patients who seroconverted after the first dose.” Figure 2 doesn't fit with the response seen in the 23 patients with a median of 1679 BAU/ml (the figure goes until 40 BAU/ml). Please, again, explain and correct.
response : the figure mentioned the medians of the 79 patients in the different categories of disease. The median of 1679 BAU/mL corresponds to the subgroup of patients who seroconverted after the first dose and were amplified after the 2nd dose.
-Lines 235-237: “To our knowledge, only 52 patients with hematological malignancies received TC, with 3.8% of patients with COVID-236 19 with a follow-up of 79 days (25).”
This statement is far away from reality. Many more oncohematologic patients have received tixagevimab/cilgavimab. Read, for example, a recent review of the topic (Akinosoglou K. Tixagevimab/Cilgavimab in SARS-CoV-2 Prophylaxis and Therapy: A Comprehensive Review of Clinical Experience. Viruses. 2023;15(1):118)
response : Sentence corresponding to this reference was introduced in the discussion section in addition to the reference.
Round 2
Reviewer 2 Report (Previous Reviewer 2)
The authors have corrected the previous commented problems.
This manuscript is a resubmission of an earlier submission. The following is a list of the peer review reports and author responses from that submission.
Round 1
Reviewer 1 Report
Abstract
According to the guidelines for authors, the Abstract should contain the purpose of the study (1) Background: Place the question addressed in a broad context and highlight the purpose of the study).
Introduction
The concepts of lines 34 to 38 can be synthesized to a single.
Patients with low levels of gammaglobulin levels (<5g/L) had reduced seroconversion (p=0.019). Compared to the group with levels higher than 5g/L?
Methods
Please could you clarify, What was the study design? In the statistical analysis paragraph, it says that it was observational, which one do you mean?
Why was the follow-up of the patients done from 2021 January till 2022 June and the ethics committee approved the project on 21 December 21, 2021.
Why was written informed consent requested in advance from patients when the project had apparently not yet been authorized?
Figure 1 can be deleted, it provides little information to the main message of the study; it is sufficient to explain that the patients were followed up with Thess® monitoring.
If the main objective of the study was: to report real-world medical follow-up and decision in a series of 216 patients with HM vaccinated with Pfizer-BioNT162b2 mRNA COVID-19 at a single institution; why are two antibody detection techniques compared? In my opinion such a situation is not part of this study. In the same manner, there is no hypothesis testing to determine concordance.
What did the clinical follow-up of the patients consist of? Only the bioclinical follow-up is detailed in the methods.
What was the reason for including a group of health care workers as a comparison group? If so, the study design would not only be observational, but analytical.
Statistical analysis
The Wilcoxon-Mann-Whitney test to establish the difference in dispersion of data from one group with respect to another; it is not useful for comparing medians.
Results
If the patients were followed up for one year, why are only the baseline results and the results of the first and second doses of the vaccine presented? I suggest adding the frequency of seroconversion to the first and second doses according to the type of disease, as well as the antibody concentrations in each of the measurements taken until the year of evaluation (cohort study).
Table 2 is not very clear; it does not have headings to identify the comparison variables.
Table 3 shows the results of the comparison of two techniques to quantify anti-SARS-Cov-2 antibodies. In my opinion, they are not part of the objective of the study (to report real-world medical follow-up and decision in a series of 216 patients with HM vaccinated with Pfizer-BioNT162b2 mRNA COVID-19 at a single institution).
The following statement suggests that the study design was analytical: "For comparison, the median serum levels of anti-spike antibodies after the 2nd dose of vaccination in the 228 patients were significantly lower than observed in 15 healthcare workers, respectively 59 BAU/mL
In my opinion, factors associated with non-seroconversion should have been identified in patients with malignant neoplasms.
Author Response
Thank you for these comments.
We edit the Abstract in order to highlight the purpose of this study and the context.
"Hematological malignancies (HM) have a heterogeneous serological response after vaccination due to disease or treatment. The aim of this real-world study was to analyze it after Pfizer-BioNT162b2 mRNA vaccination in 216 patients followed up for 1 year. The first 43 patients had an initial follow-up by telemedicine (TM) system with no major event reported. The anti-Spike IgG antibodies were checked 3-4 weeks post-1st vaccination and every 3-4 months, by 2 standard bioassays and a rapid serological test (RST). Vaccine boosts were given when the level was <7BAU/mL. Patients who did not seroconvert after 3-4 doses received tixagevimab/cilgavimab (TC). 15 results were discordant between 2 standard bioassays. Good agreement was observed between standard and RST on 97 samples. After 2 doses, 68% were seroconverted (median=59BAU/mL) with a median of 162BAU/mL and 9BAU/mL, respectively in untreated and treated patients (p<0.001), particularly for patients receiving rituximab. Patients with gammaglobulin levels <5g/L had reduced seroconversion compared to higher levels (p=0.019). Median levels were 228BAU/mL post-2nd dose if seroconverted post-1st and 2, if seroconverted only post-2nd dose. 68% of post-2nd dose negative patients were post-3rd dose positive. 16% received TC, 6 with non-severe symptomatic COVID-19 within 15-40 days. Personalized serological follow-up should apply particularly to HM patients."
This study is an observational study of real-world data based on one clinician's experience and decision-making, as defined by in the article from Makady A et al. Value in health 2017; 20: 858-65. In addition, we have already gained experience with vaccination against H1N1 in patients with hematological malignancies and this has encouraged us to quickly monitor the serological response in this difficult time of COVID-19 with all the initial uncertainties.
The concepts of lines 34 to 38 have been synthesized in one sentence:
Response : "Vaccination represents the best validated strategy against the SARS-Cov-2 pandemic, especially for high-risk patients, such as patients with hematological malignancies having a delayed or weak specific immune response after standard vaccination due to the disease or a treatment (1-3)."
Reviewer comment : Patients with low levels of gammaglobulin levels (<5g/L) had reduced seroconversion (p=0.019). Compared to the group with levels higher than 5g/L?
Response: Reduced seroconversion was significantly lower between patients with low of gammaglobulin levels(<5g/L) than those with higher levels >5g/L.
Reviewer Comment : Why was the follow-up of the patients done from 2021 January till 2022 June and the ethics committee approved the project on 21 December 21, 2021.
Response : There was a typo for the date of the internal ethics committee : it was December 21, 2020 before the start of collecting the data.
Figure 1 and its comment have been deleted.
The question of the usefulness of the 2 biological tests for serological detection is raised. In this medical follow-up in real situation, at the beginning of the vaccination period, we had an objective to clearly define the serological response. However, due to the lack of comparison and standard Unit of the antibody activity, we used the 2 main techniques. Thus, our laboratory doubled the serological follow-up and the comparison could be made. Moreover, this comparison of 2 standard techniques also made it possible to have an evaluation of a less expansive rapid test in the event of a personalized follow-up which is suggested. We thought that this assessment had additional scientific interest for this article. This comparison was merely descriptive in a real-life situation where the choice of test involved a certain difficulty, thus making it possible to establish the scientific basis of the interpretation and the interpretability for the article and further publications. They do not correspond to a test hypothesis.
We modified the text to better explain this aspect:
line 87: "Due to the lack of a biological standard at the start of serological analyses, the central laboratory uses a second bioassay performed in another laboratory from the same group (LABOSUD/INOVIE, Montpellier, France)." and the sentence "Each sample was tested in duplicate with the 2 assays, by 2 independent labs (included in the group LABOSUD/INOVIE, Montpellier, France)." was suppressed.
Question: What did the clinical follow-up of the patients consist of? Only the bioclinical follow-up is detailed in the methods.
Response: The patients were followed clinically and biologically by a single investigator, including standard follow-up and treatment decisions. We add :
"...standard clinical and biological monitoring, …», line 81.
Question: What was the reason for including a group of health care workers as a comparison group? If so, the study design would not only be observational, but analytical.
Response: We agree that this group was not of great interest in this observational study. This group was set up at the start to situate the serological level of the patients in a somewhat blind initial context. The laboratory also wanted to situate the level of responses of these patients.
Comment: "The Wilcoxon-Mann-Whitney test to establish the difference in dispersion of data from one group with respect to another; it is not useful for comparing medians."
Response: We agree that Wilcoxon-Mann-Whitney test was not useful in this statistical analysis and we suppress it.
Comment: If the patients were followed up for one year, why are only the baseline results and the results of the first and second doses of the vaccine presented? I suggest adding the frequency of seroconversion to the first and second doses according to the type of disease, as well as the antibody concentrations in each of the measurements taken until the year of evaluation (cohort study).
Vaccination recommandations from Pfizer and the French government were 2 doses. (HAS 10/12/20 https://www.has-sante.fr/upload/docs/application/pdf/2020-12/strategie_de_vaccination_contre_le_sars-cov-2_-_recommandations_intermediaires_sur_les_modalites_de_mise_en_oeuvre_de_la_vac.pdf ; https://www.has-sante.fr/jcms/p_3234097/fr/modification-du-schema-vaccinal-contre-le-sars-cov-2-dans-le-nouveau-contexte-epidemique; https://ansm.sante.fr/tableau-vaccin/comirnaty-pfizer-biontech
Considering the context of this observational study and the recommandations for 2 doses, we analyzed only 79 patients after the first vaccination.
We introduced this analysis in the following sentence:
« 79 patients had serological analysis both after the first and second dose of vaccine. As seen in Fig.2, the median levels of anti-spike Ab were amplified after the second dose, with better amplification for CLL patients. Of these 79 patients, 58 were not seroconverted after the first dose, with only 24/58 seroconverted after the second dose with a median anti-S antibody at 16 BAU/ML (range 1-270) compared to a median antibody at 1679 BAU/mL (range 549-5516) after the 2nd dose for the 23 patients who seroconverted after the first dose."
Table 2 is not very clear; it does not have headings to identify the comparison variables.
Response : Table 2 was changed with headings
|
Seroconverted after 2 doses |
Quant® Assay (Roche) |
|
|||
|
NO |
YES |
Total |
P<0.001 |
||
|
Elecsys Quant® Assay (Abbott) |
NO |
42 (23%) |
12 (6.6%) |
54 (30%) |
|
|
YES |
3 (1.6%) |
125 (69%) |
128 (70%) |
||
|
Total |
45 (25%) |
137 (75%) |
182 (100%) |
||
Table 3 shows the results of the comparison of two techniques to quantify anti-SARS-Cov-2 antibodies. In my opinion, they are not part of the objective of the study (to report real-world medical follow-up and decision in a series of 216 patients with HM vaccinated with Pfizer-BioNT162b2 mRNA COVID-19 at a single institution).
Response: As mentioned earlier, at the start of this observational study, the Units were heterogenous and company-dependent. Thus, we used 2 standard tests and found few differences. This encourages us to continue the comparison. In this article, we relate the reflection and decisions along the way in the described context of real-life medicine that could lead to a more personalized follow-up with a less expensive rapid test. Thus, the comparison with the 2 tests used could make sense and complete the message and the originality. We are aware that this article suggest more than demonstrates such an attitude.
The following statement suggests that the study design was analytical: "For comparison, the median serum levels of anti-spike antibodies after the 2nd dose of vaccination in the 228 patients were significantly lower than observed in 15 healthcare workers, respectively 59 BAU/mL
In my opinion, factors associated with non-seroconversion should have been identified in patients with malignant neoplasms.
Response: We agree with this comment and suppress the comparison with 15 healthcare workers.
Discussion section was modified.
Reviewer 2 Report
This is a small unicentric study focused on the serologic response of SARS-CoV-2 vaccination in a mixed hematologic group of 228 patients.
The main problems with the manuscript are:
-Unclear aim. What is the message/s of this study?
-A hypothesis that is not answered, and can’t be answered, with the data of this manuscript.
-Repeated numeric inconsistencies
-English should be reviewed
-Confused results section, difficult to understand.
a) Abstract
- What do you mean by “Tolerance using TM was good”? I understand that the tolerance to the vaccine was well evaluated by TM. Please rewrite the sentence.
b) Introduction
1.-Page 2, lines 46-52 “The recommendations for the systematic use of vaccine boosts in COVID-19 for the entire population or subgroups considered to be at risk is a non-personalized attitude and may increase toxicity or be useless for the objective set by achieving immune overstimulation which must be evaluated. A more reasonable medical policy could be based on patient follow-up both with a telemedicine application for tolerance during the vaccination period and longitudinal follow-up of seroconversion with rapid serological tests (RST) to control a more personalized vaccination follow-up”
This is a hypothesis that this study doesn't answer. The objective of vaccination is to prevent infection or/and severe forms of COVID-19. Tolerance to SARS-CoV-2 has been analyzed in several studies finding that, in general, their use is safe in oncohematologic patients. A small study of 228 is unlikely to change what is known about vaccine tolerance.
A personalized vaccination, as commented by the authors, should be based on the analysis of the previous objectives commented (prevention of infection or/and severe forms of COVID-19.), and this was not done. Moreover, with a small group of mixed cases, it is not possible to achieve this. So, a different rationale for the study should be given.
c) Methods
1-Page 3, lines 91-92: “A serological follow-up was regularly carried out to define the medical policy, ..”
What was that policy? Please, give the details of how third vaccine doses were administered based on the serologic results. It is not clear what the criteria were for the third (64% of the patients) and fourth (6.1% of patients) vaccine doses.
2.-Figure 1 doesn't add interesting information to the manuscript. I would delete it.
d)Results
1.-It is not necessary to repeat all the numbers of the underlying diseases in the text and in table 1.
2.-Mismatches between text and table 1
-In the text, it is said that the median age for the 228 patients was 73 years (range 65-79), but in table 1 appears as 74 (62-82). Please, correct and explain.
-105 patients (46%) received concomitant therapy for their disease, and 125 patients (54%) had no therapy during this time of feedback. That sum to 230 patients, but the total number of patients was 228. Please, correct this.
3.-Table 1
-there is a “*” for the number 4. There is no legend to explain what means this “*”. Please, correct/explain.
-The sum of only 2 doses, only 3 doses, and 4 doses is 230, but the total number of patients was 228. Please correct.
-The sum of patients with 3 doses is 144 and not 146 as appears in the table. Please correct.
-The sum of patients that seroconverted after the second dose sum 147 and not 145 as appears in the table.
-The levels of anti-S ab that appears, I suppose are after the second dose. Add this information in the title of that raw
-Erroneous percentages:
-146 patients received a third dose, which represents 64% of the 228 patients and not 68% as appears in the table.
-145 patients appear as seroconverted after the second dose, which is 63.5% and not 68% as appears in the table.
4.-Early assessment of tolerance post-vaccination follow-up by telemedecine.
-Line 131: “If transient, within a day, local pain was common, especially after the second dose.”
What do you want to say? I can’t understand the meaning of this sentence.
5.-Table 2. In theory, this table is for the comparison of seroconversion of 2 tests, but only one set of data is given. The data from both tests should appear separately.
6-Table 3 is not clear to me. For example, what means 11/52+ in the column of Quant® Assay? For Elecsys appear “8/11”; what mean?
7.-Seroconversion levels
-“As shown in Table 3, after the second dose of vaccine, respectively 68% of the patients 147 with HM were seroconverted with a median level of antibody at 58 BAU/mL (range 3-148 319).”
Looking at table 3, I can’t find the 68% seroconversion rate nor the median level of antibodies of 58 BAU/ml.
-“The median of the anti-Spike Ab was 228 BAU/mL 150 (range 78-783) for the patients who were seronconverted after the first dose in comparison to a median of 2 BAU/mL (range 0.1-38) for the patients who were not seroconverted after the first dose, corresponding to only 43% of the patients who had seroconversion after the second dose if they were negative after the first vaccination”
-I can’t understand what you mean, particularly “corresponding to only 43% of the patients who had seroconversion after the second dose if they were negative after the first vaccination”
-What is the aim of comparing the title of those who seroconverted and those who not?
-What serologic technique was used for these results (seroconverted vs non-seroconverted)?
-In the text, it is said that the median anti-Spike Ab was 228 BAU/mL (range 78-783) for the patients who were seroconverted, but in figure 2, the scale goes only up to 40 BAU/ml
8.-Factors influencing seroconversion.
“The CMV status based on the presence of anti-CMV IgG antibodies was known in 36 patients, but with no difference between the 2 subgroups”
To which subgroups are you referring? Moreover, with only 36 patients, it is difficult to find a significant difference unless the difference is huge.
9.-Bio-medical follow-up and efficacy.
-14 patients did not seroconvert, but 16 receive Evusheld. Explain/correct.
10.-Discussion
-With only 43 patients assessed for tolerance, not much can be said about the safety of the SARS-CoV-2 vaccines.
-The discussion is too vague and long, and not centred on the results of the study. Many different aspects are discussed without data from the study, like variants of SARS-CoV-2 and vaccine efficacy.
Author Response
Reviewer comment: "This is a small unicentric study focused on the serologic response of SARS-CoV-2 vaccination in a mixed hematologic group of 228 patients."
Response: We agree that it is a unicentric study, however with consistant number of patients, similarly than most recent studies published in this domain (Pascale SP et al Front Immunol 2022 Aug 8;13:892331; with 215 patients and Ollila TA et al Cancer 2022 Sep 15;128(18):3319-3329, with 181 patients).
Reviewer comment : The main problems with the manuscript are:
-Unclear aim. What is the message/s of this study?
-A hypothesis that is not answered, and can’t be answered, with the data of this manuscript.
Responses: This is a single-center observational study, with a single investigator testifying to homogeneity of thought and action and this represents the originality of this study comparing to the literature. Thus, the first message shows that in real life, there is indeed a heterogeneity of serological response impacted by the therapies confirming data from analytical or prospective studies. The second message is a strong suggestion for a more personalized follow-up which is not proposed by the authorities who recommend systematic vaccinations for all populations. These patients with hematological malignancies may require such an approach and tools to achieve this. The substance of this article thus relates an attitude of real medical life and brings an element of reflection for possible prospective studies like a pilot study. It is in that this study is both original.
-Repeated numeric inconsistencies
Response: We change these numeric inconsistencies as discussed elsewhere.
-English should be reviewed
Response: English has been reviewed
-Confused results section, difficult to understand.
Response: We clarified the results section.
What do you mean by “Tolerance using TM was good”? I understand that the tolerance to the vaccine was well evaluated by TM. Please rewrite the sentence.
Response: We suppress this sentence in the abstract and modify the following sentence:
"The first 43 patients had an initial follow-up by telemedicine (TM) system with no major event reported. "
Reviewer comment:
- b) Introduction
1.-Page 2, lines 46-52 “The recommendations for the systematic use of vaccine boosts in COVID-19 for the entire population or subgroups considered to be at risk is a non-personalized attitude and may increase toxicity or be useless for the objective set by achieving immune overstimulation which must be evaluated. A more reasonable medical policy could be based on patient follow-up both with a telemedicine application for tolerance during the vaccination period and longitudinal follow-up of seroconversion with rapid serological tests (RST) to control a more personalized vaccination follow-up”
This is a hypothesis that this study doesn't answer. The objective of vaccination is to prevent infection or/and severe forms of COVID-19. Tolerance to SARS-CoV-2 has been analyzed in several studies finding that, in general, their use is safe in oncohematologic patients. A small study of 228 is unlikely to change what is known about vaccine tolerance.
A personalized vaccination, as commented by the authors, should be based on the analysis of the previous objectives commented (prevention of infection or/and severe forms of COVID-19.), and this was not done. Moreover, with a small group of mixed cases, it is not possible to achieve this. So, a different rationale for the study should be given.
Response : This study is not intended to demonstrate any tolerance of the vaccine or the benefit of personalized vaccination. We agree with the reviewer that the objective of vaccination is to prevent infection and/or severe forms of COVID-19. Of note, in this series, only 39 patients (17%) presented symptomatic COVID-19 with no patient in ICU. Thus, in a real-life study, insofar as the benefit of a vaccine is clearly observed, certain suggestions for action to be taken, can be formulated, all the more so if they have not been mentioned in the literature.
Comment: 1-c) Methods
1-Page 3, lines 91-92: “A serological follow-up was regularly carried out to define the medical policy, ..”
What was that policy? Please, give the details of how third vaccine doses were administered based on the serologic results. It is not clear what the criteria were for the third (64% of the patients) and fourth (6.1% of patients) vaccine doses.
Response: The adopted policy was defined from the 2nd dose. A third dose was administered if the patient was not seroconverted or if the level of antibody was below 10BAU/ml. (Quant® Assay, Abbott). For a fourth dose, if the patient had never been seroconverted, we favored the use of TC or TI.
We modified the sentence " . A serological follow-up was regularly carried out to define the medical policy, including a third dose for all patients with a decrease in the seroconversion level and a fourth dose for some of them" and replace it by the following one : "A serological follow-up was performed as mentioned elsewhere with a third dose administered if the patient was not seroconverted or if the level of antibody was below 10BAU/ml. (Quant® Assay, Abbott). For a fourth dose, if the patient had never been seroconverted, we favored the use of TC or TI. «
2.-Figure 1 doesn't add interesting information to the manuscript. I would delete it.
Response: I agree and we delete it.
d)Results
1.-It is not necessary to repeat all the numbers of the underlying diseases in the text and in table 1.
Response: It has been changed with only mentioned in the text with the details of lymphoma patients (NHL and HD)
2.-Mismatches between text and table 1
-In the text, it is said that the median age for the 228 patients was 73 years (range 65-79), but in table 1 appears as 74 (62-82). Please, correct and explain.
Response: Median age and range has been recalculated and for the total it is 73 (range 63-82). This was due to the first version of the article with some patients lacking who were then integrated in the analysis.
-105 patients (46%) received concomitant therapy for their disease, and 125 patients (54%) had no therapy during this time of feedback. That sum to 230 patients, but the total number of patients was 228. Please, correct this.
Response: This was recalculated : 105 patients treated and 123 untreated.
3.-Table 1
-there is a “*” for the number 4. There is no legend to explain what means this “*”. Please, correct/explain.
Response: I apologize for these errors
4 doses
-The sum of only 2 doses, only 3 doses, and 4 doses is 230, but the total number of patients was 228. Please correct.
Response: done
-The sum of patients with 3 doses is 144 and not 146 as appears in the table. Please correct.
Done
-The sum of patients that seroconverted after the second dose sum 147 and not 145 as appears in the table.
Done
-The levels of anti-S ab that appears, I suppose are after the second dose. Add this information in the title of that raw
Done
-Erroneous percentages:
-146 patients received a third dose, which represents 64% of the 228 patients and not 68% as appears in the table.
Response : Recalculation 144 patients representing 63%
-145 patients appear as seroconverted after the second dose, which is 63.5% and not 68% as appears in the table.
Response : recalculation 147 patients (64%)
4.-Early assessment of tolerance post-vaccination follow-up by telemedecine.
-Line 131: “If transient, within a day, local pain was common, especially after the second dose.”
What do you want to say? I can’t understand the meaning of this sentence.
Response: the sentence was changed: "Local pain at the injection site, generally transient, was mainly reported after the second dose. "
5.-Table 2. In theory, this table is for the comparison of seroconversion of 2 tests, but only one set of data is given. The data from both tests should appear separately.
Response: Table 2 represents the concordance of the positivity and negativity between the 2 tests. We modified the Table and the title for better understanding.
Vaccination recommandations from Pfizer and the French government were 2 doses. (HAS 10/12/20 https://www.has-sante.fr/upload/docs/application/pdf/2020-12/strategie_de_vaccination_contre_le_sars-cov-2_-_recommandations_intermediaires_sur_les_modalites_de_mise_en_oeuvre_de_la_vac.pdf ; https://www.has-sante.fr/jcms/p_3234097/fr/modification-du-schema-vaccinal-contre-le-sars-cov-2-dans-le-nouveau-contexte-epidemique; https://ansm.sante.fr/tableau-vaccin/comirnaty-pfizer-biontech
Considering the context of this observational study and the recommandations for 2 doses, we analyzed only 79 patients after the first vaccination and after the second dose.
Table 2 is not very clear; it does not have headings to identify the comparison variables.
Response : Table 2 was changed with headings and title.
"Concordance of the seroconversion after 2 doses between Elecsys..."
|
Seroconverted after 2 doses |
Quant® Assay (Roche) |
|
|||
|
NO |
YES |
Total |
P<0.001 |
||
|
Elecsys Quant® Assay (Abbott) |
NO |
42 (23%) |
12 (6.6%) |
54 (30%) |
|
|
YES |
3 (1.6%) |
125 (69%) |
128 (70%) |
||
|
Total |
45 (25%) |
137 (75%) |
182 (100%) |
||
6-Table 3 is not clear to me. For example, what means 11/52+ in the column of Quant® Assay? For Elecsys appear “8/11”; what mean?
Response : I agree and a typo was introduced and needed to be explained. 52/97 patients tested with RST were negative but 11 among these 52 patients were positive with Quant Assay and 8/52 (not 11, the typo) were positive with Eleecsys. However, these 11 and 8 false negative had low titers (mean and range).
We modified and corrected this Table 3. We add in the legend : "11/52 patients and 8/11 were positive, respectively with Quant Assay and Elecsys Assay, considered as false negative with RST, however with low titers (median, range). "
7.-Seroconversion levels
-“As shown in Table 3, after the second dose of vaccine, respectively 68% of the patients 147 with HM were seroconverted with a median level of antibody at 58 BAU/mL (range 3-148 319).”
Looking at table 3, I can’t find the 68% seroconversion rate nor the median level of antibodies of 58 BAU/ml.
Response: I apologize it is in Table 1, with corrected numbers: Sentence was changed.
"As shown in Table 1, after the second dose of vaccine, 64% of the 228 patients were seroconverted with a median level of antibody at 59 BAU/mL (range 0-1889), including 62% of the 216 patients with HM (median 55 BAU/mL, range 0-896)."
Questions:
-I can’t understand what you mean, particularly “corresponding to only 43% of the patients who had seroconversion after the second dose if they were negative after the first vaccination”
-What is the aim of comparing the title of those who seroconverted and those who not?
-What serologic technique was used for these results (seroconverted vs non-seroconverted)?
Response : ABBOTT.
-In the text, it is said that the median anti-Spike Ab was 228 BAU/mL (range 78-783) for the patients who were seroconverted, but in figure 2, the scale goes only up to 40 BAU/ml
Response:
"79 patients had serological analysis both after the first and second dose of vaccine. As seen in Fig.2, the median levels of anti-spike Ab were amplified after the second dose, with better amplification for CLL patients. Of these 79 patients, 58 were not seroconverted after the first dose, with only 24/58 seroconverted after the second dose with a median anti-S antibody at 16 BAU/ML (range 1-270) compared to a median antibody at 1679 BAU/mL (range 549-5516) after the 2nd dose for the 23 patients who seroconverted after the first dose."
Fig.2 expresses the curves of the median values of the anti-spike antibodies for the 79 patients who dosages after the 1st and the second dose. Thus, those median values are under 40BU/mL.
We modified the figure with "median" mentioned.
8.-Factors influencing seroconversion.
“The CMV status based on the presence of anti-CMV IgG antibodies was known in 36 patients, but with no difference between the 2 subgroups”
To which subgroups are you referring? Moreover, with only 36 patients, it is difficult to find a significant difference unless the difference is huge.
Response: At the time of writing the article, we only had 36 patients tested for CMV moment we had only 36 patients tested for CMV. The status status was analyzed retrospectively on frozen serum at the same time as the dosage Anti-S Ab post 2nd dose, for 85 patients. 64 were positive (all with only IgG CMV) and 21 were negative. Median IgG Ab were 5.7 BAU/mL (range 0-1066) for the 64 patients and 4.6 BAU/mL (range 0-39 634).
9.-Bio-medical follow-up and efficacy.
-14 patients did not seroconvert, but 16 receive Evusheld. Explain/correct.
Response: 14 patients were not seroconverted and 2 additional patients who had very low Anti-S titers Ab (less than 10BAU/mL with Abbott test) received TC. We modified the text.
10.-Discussion
-With only 43 patients assessed for tolerance, not much can be said about the safety of the SARS-CoV-2 vaccines.
Response:
-The discussion is too vague and long, and not centred on the results of the study. Many different aspects are discussed without data from the study, like variants of SARS-CoV-2 and vaccine efficacy.
Responses:
We modified and shortened the Discussion section.
We introduced the TC discussion in the context of HM which was not well documented in the literature (Sturver R et al Cancer Cell 2022 Jun 13;40(6):590-591. doi: 10.1016/j.ccell.2022.05.007.) from MSKCC experience in 52 patients.
"In the randomized PROVENT TC study, only 7% of the participants had cancer or cancer history (Levin MJ et al . NEJM 2022). To our knowledge, only 52 patients with hematological malignancies received TC, with 3.8% of patients with COVID-19 with a follow-up of 79 days (Sturver R et al Cancer Cell 2022). Thus, this real-life study integrates a complete medical strategy adapted to high-risk patients with heterogeneous seroconversion, in particular the use of TC with an infection rate of 17% but no serious form.